# TNF-α and IL-1β Modulate Blood-Brain Barrier Permeability and Decrease Amyloid-β Peptide Efflux in a Human Blood-Brain Barrier Model

**DOI:** 10.3390/ijms231810235

**Published:** 2022-09-06

**Authors:** Romain Versele, Emmanuel Sevin, Fabien Gosselet, Laurence Fenart, Pietra Candela

**Affiliations:** 1Laboratoire de la Barrière Hémato-Encéphalique (LBHE), UR 2465, Université d’Artois, F-62300 Lens, France; 2Endocrinology, Diabetes and Nutrition Unit, Institute of Experimental and Clinical Research, Medical Sector, Université Catholique de Louvain, 1200 Brussels, Belgium

**Keywords:** blood-brain barrier, Alzheimer’s disease, amyloid-β peptide, inflammation, TNF-α, IL-1β

## Abstract

The blood-brain barrier (BBB) is a selective barrier and a functional gatekeeper for the central nervous system (CNS), essential for maintaining brain homeostasis. The BBB is composed of specialized brain endothelial cells (BECs) lining the brain capillaries. The tight junctions formed by BECs regulate paracellular transport, whereas transcellular transport is regulated by specialized transporters, pumps and receptors. Cytokine-induced neuroinflammation, such as the tumor necrosis factor-α (TNF-α) and interleukin-1β (IL-1β), appear to play a role in BBB dysfunction and contribute to the progression of Alzheimer’s disease (AD) by contributing to amyloid-β (Aβ) peptide accumulation. Here, we investigated whether TNF-α and IL-1β modulate the permeability of the BBB and alter Aβ peptide transport across BECs. We used a human BBB in vitro model based on the use of brain-like endothelial cells (BLECs) obtained from endothelial cells derived from CD34+ stem cells cocultivated with brain pericytes. We demonstrated that TNF-α and IL-1β differentially induced changes in BLECs’ permeability by inducing alterations in the organization of junctional complexes as well as in transcelluar trafficking. Further, TNF-α and IL-1β act directly on BLECs by decreasing LRP1 and BCRP protein expression as well as the specific efflux of Aβ peptide. These results provide mechanisms by which CNS inflammation might modulate BBB permeability and promote Aβ peptide accumulation. A future therapeutic intervention targeting vascular inflammation at the BBB may have the therapeutic potential to slow down the progression of AD.

## 1. Introduction

The blood-brain barrier (BBB) is a dynamic structure whose anatomical seat is the endothelial cells of the brain capillaries, at the interface between the blood compartment and the central nervous system (CNS). This important physiological barrier is required for brain homeostasis and protection of the sensible neuronal environment. Barrier functions of the BBB are conferred by specialized features of brain endothelial cells (BECs), presenting a low rate of transcytosis as well as a strongly limited paracellular permeability, thus preventing the unspecific transport of various substances between blood and CNS. The major factor limiting BBB permeability is the presence of an elaborate network of junctional complexes, consisting of tight junctions (TJs) and adherent junctions (AJs). Occludin, tricellulin, zonula occludens-1 (ZO-1), claudin-3 and claudin-5 are key components of TJs, whereas AJs are mainly represented by vascular endothelial-cadherin (VE cadherin). Their level of expression, as well as their localization in the BECs, are important factors in maintaining the physical barrier property and thus preventing paracellular diffusion [1]. BBB permeability is altered in many CNS pathologies and is an early and prominent pathological feature of several neuroinflammatory diseases, including Alzheimer’s disease (AD) [1].

AD is a neurodegenerative disorder characterized by slow, progressive and degenerative changes, the main feature of which is the accumulation of amyloid-β (Aβ) peptide in the brain and within the walls of intracerebral microvessels [2,3]. For several years, compelling evidence has demonstrated that the BBB has an active role in the onset and development of AD [3]. Indeed, the BBB physiologically regulates the influx (blood-to-brain) and the efflux (brain-to-blood) of Aβ peptides through the presence of different receptors and transporters, such as receptor for advanced glycation endproducts (RAGE), P-glycoprotein (P-gp), breast cancer resistance protein (BCRP) and low-density lipoprotein receptor-related protein 1 (LRP1), as well as phosphatidylinositol binding clathrin assembly (PICALM) [4,5,6,7,8,9]. Whereas RAGE mediates Aβ peptide entrance from the blood [4,5], LRP1 is mainly involved in Aβ peptide efflux from the brain [6]. P-gp and BCRP also mediate this efflux, but restrict Aβ peptide entrance [5,7]. More recently, PICALM has been described as being involved in brain Aβ peptide efflux mediated by the LRP1-P-gp transport pathway [8,9]. Numerous reports have indicated that dysregulation of their expression and functionality are triggers or contributors to AD [8,10,11,12,13]. However, although these mechanisms are endorsed by the scientific community, the causes of these changes remain unexplained.

New data assumed that neuroinflammation triggered by proinflammatory cytokines contributes to BBB dysfunction and is a key component of the pathological lesions found in the brains of AD patients [3,14,15]. Two of these proinflammatory mediators are tumor necrosis factor-α (TNF-α) and another relevant, less studied cytokine, interleukin-1β (IL-1β), for which the presence around Aβ plaques have been reported in the postmortem brain tissue of both transgenic AD mice and AD patients [16]. Although a number of in vivo studies have shown that TNF-α and IL-1β are able to modify BBB properties [14,15,17,18], the effects of these proinflammatory cytokines on endothelial dysfunction and their consequences on Aβ peptide transport has been less studied in the literature. In addition, there have been many studies conducted on in vitro models to evaluate “BBB modulation” in response to an inflammatory stimulus. Few of them use a human model of the BBB based on primary cultured cells, which is essential to recapitulate many of the immune functions of the BBB that are known to occur in vivo [19]. Therefore, to address this issue, this study was designed to investigate whether TNF-α and IL-1β (to mimic CNS inflammation) can modulate the BBB properties and contribute to the development of AD pathology by altering Aβ peptide transport. For this purpose, we used the patented human BBB in vitro model (i.e., the brain-like endothelial cells model—BLECs) developed in the laboratory and displaying BBB properties closed to those observed in vivo [20,21].

## 2. Results

### 2.1. BLEC Response to TNF-α and IL-1β Treatment

To mimic CNS inflammation, tumor necrosis factor-α (TNF-α) or interleukin-1β (IL-1β) were added at 10 ng·mL^−1^ for 24 h to the abluminal compartment of the BBB model (corresponding to the brain side).

An MTT assay was performed as a preliminary test to determine their effect on BLEC viability and to validate the chosen concentrations. Dimethyl sulfoxide (DMSO) at 10% (*v*/*v*) was used as a positive control of cell toxicity. As shown in Figure 1A, after 24 h of exposure to 10% DMSO, BLEC viability was decreased (by 26.5%, *p*-value = 0.0003) compared to control. By contrast, exposure to TNF-α and IL-1β at 10 ng·mL^−1^ for 24 h did not affect BLEC viability, indicating that the two inflammatory molecules did not induce cell death at this concentration.

Next, as the inflammatory response of endothelial cells is largely dependent on appropriate upregulation of adhesion molecules, such as intercellular adhesion molecule-1 (ICAM-1) and vascular cell adhesion molecule-1 (VCAM-1) [3,22,23], and the inflammasome molecule NOD-like receptor family, pyrin domain containing 3 (NLRP3) [24], we evaluated, by Western immunoblotting, the ability of BLECs to respond to these pro-inflammatory stimuli. We showed that the protein levels of ICAM-1, VCAM-1 and NLRP3 were absent or low in untreated BLECs but became significantly higher following TNF-α or IL-1β treatment (Figure 1B). Interestingly, at an equivalent concentration, TNF-α generates a stronger inflammatory response than IL-1β as observed from the Western immunoblotting study, where the levels of ICAM-1, VCAM-1 and NLRP3 proteins were higher after exposure with TNF-α compared to with IL-1β by 22.2% (*p*-value = 0.05), 230.6% (*p*-value < 0.0001) and 478.4% (*p*-value < 0.0001), respectively (Figure 1C). Overall, these results demonstrate that proinflammatory cytokine treatments do not alter cell viability but can induce inflammatory response in BLECs.

### 2.2. Effect of TNF-α and IL-1β on BBB Permeability

Next, we examined the effect of TNF-α and IL-1β on BBB permeability. The physical integrity of the BLECs was monitored by measuring the rate of passage of the small hydrophilic marker lucifer yellow (LY; ~400 Da) from the upper to the lower compartment to calculate the endothelial permeability coefficient of LY (Pe_LY_). BLECs treated with TNF-α and IL-1β for 24 h showed a significant increase in Pe_LY_ values from 0.54 ± 0.03 × 10^−3^ cm·min^−1^ to 1.00 ± 0.03 × 10^−3^ cm·min^−1^ (*p*-value < 0.0001) and to 0.95 ± 0.05 × 10^−3^ cm·min^−1^ (*p*-value < 0.0001), respectively, compared to untreated BLECs. These results showed that the permeability of BLECs was altered in inflammatory conditions (Figure 2A).

The alteration of BLEC permeability induced by TNF-α and IL-1β is not linked with BLEC cell death, as demonstrated in Figure 1A. Therefore, we suggest that it can be attributed to alterations in the AJ and TJ, thus modifying transcellular or paracellular routes of molecules [14,15]. To discriminate the impact of TNF-α and IL-1β on these different pathways and based on our previous Pe_LY_ results obtained at 37 °C informing on the global crossing of Pe_LY_ (paracellular and transcellular), a Pe_LY_ experiment was also performed at 4 °C. As we previously demonstrated, all active mechanisms are inhibited at 4 °C, thus stopping the transcellular routes of BLECs [5]. Thus, inhibition of the transcellular transport allows us to observe only the modification on the paracellular permeability. Then, the transcellular transport of LY, corresponding to the difference between global Pe_LY_ and Pe_LY_ at 4 °C, was determined. As shown in Figure 2B, TNF-α treatment significantly increased the transcellular and paracellular passages of LY through the BLEC monolayer by 40.1% (*p*-value < 0.0001) and 175.2% (*p*-value = 0.0022), respectively, compared to the control. Similarly, IL-1β treatment also significantly increased the transcellular and paracellular passage of LY through the BLEC monolayer by 94.2% (*p*-value < 0.0001) and 41.3% (*p*-value = 0.0152), respectively, compared to the control. Furthermore, if we compare the effect of the cytokines on BBB permeability between them, our study showed a greater effect of IL-1β exposure on the increase in transcellular passage of LY through the BLEC monolayer compared to TNF-α treatment (+ 54.2%, *p*-value < 0.0001). In contrast, a greater increase in paracellular passage of LY through the BLEC monolayer was found following TNF-α exposure compared to IL-1β treatment (+ 133.9%, *p*-value = 0.0022). These results suggest that the two tested cytokines might have a different mode of action on the parameters that regulate BBB permeability.

### 2.3. Effect of TNF-α and IL-1β on Tight Junctions at the BBB

BBB has restrictive properties primarily due to the closed junctional complex. Paracellular changes in BBB permeability induced by TNF-α and IL-1β could correlate with changes in the expression of tight junction (TJ) and adherent junction (AJ) proteins at the transcript and protein level. For this purpose, real-time qPCR and Western immunoblotting experiments were employed to evaluate the effects of TNF-α and IL-1β exposure on the expression of ZO-1, tricellulin, occludin, claudin-3 and claudin-5 as representatives of TJ proteins, as well as VE-cadherin as a representative of AJ proteins.

As shown in Figure 3A,D, the mRNA level of the VE-cadherin was not significantly different following TNF-α exposure (Figure 3A), but was lower after IL-1β treatment (Figure 3D). The mRNA levels of ZO-1, tricellulin, occludin and claudin-3 were significantly decreased compared to the control condition by 22,2% (*p*-value = 0.0065), 68.1% (*p*-value < 0.0001), 66.3% (*p*-value < 0.0001) and 61.8% (*p*-value < 0.0001), respectively, after TNF-α exposure; and by 48.1% (*p*-value < 0.0001), 62.8% (*p*-value < 0.0001), 64.0% (*p*-value < 0.0001) and 81.3% (*p*-value < 0.0001), respectively, after IL-1β exposure, compared to the control. In contrast, the expression of claudin-5 was significantly increased by 139.4% (*p*-value < 0.0001) and 48.1% (*p*-value = 0.0005) after TNF-α and IL-1β treatments, respectively.

Western immunoblotting analysis showed that VE-cadherin and ZO-1 protein levels were not significantly different after TNF-α treatment (Figure 3B), whereas ZO-1 expression slightly decreased following IL-1β exposure by 26.4% (*p*-value = 0.0022) (Figure 3E). According to the decrease in mRNA level, the protein levels of tricellulin, occludin and claudin-3 were also significantly decreased following TNF-α exposure, respectively, by 50.9% (*p*-value < 0.0001), 38.1% (*p*-value < 0.0001) and 57.3% (*p*-value < 0.0001) (Figure 3B), and following IL-1β exposure by 51.0% (*p*-value < 0.0001), 33.5% (*p*-value = 0.0009) and 73.1% (*p*-value < 0.0001) (Figure 3E), respectively. On the other hand, while Claudin-5 protein expression after TNF-α exposure increased in line with qRT-PCR analysis by 40.2% (*p*-value = 0.0021) compared to the control condition, the increase in mRNA was not reflected by a change in Claudin-5 protein levels after IL-1β exposure, as measured by immunoblotting (Figure 3B,E).

Changes in TJ protein expression induced by proinflammatory factors have often been associated with the delocalization of these proteins to cytosolic compartments [25,26,27]. We examined the localization of TJ proteins in BLECs after TNF-α and IL-1β treatment using immunofluorescence. As shown in Figure 3G, in control conditions, ZO-1, occludin, claudin-3 and claudin-5 appeared exclusively as a near-continuous staining at the cell border, as observed after IL-1β treatment. On the other hand, an increase in claudin-3 and claudin-5 cytoplasmic staining (punctuate dots) was observed after TNF-α exposure (Figure 3G). To support these observations, claudin area fraction stainings were quantified within BLECs (Figure 3H,I). A significant increase in claudin-3 area fraction staining was only observed after TNF-α treatment compared to the control condition and IL-1β treatment, at 150.2% (*p*-value = 0.021) and 129.8% (*p*-value = 0.05), respectively (Figure 3H). Similarly, only TNF-α treatment significantly increased claudin-5 area fraction staining in BLECs by 161.8% (*p*-value < 0.001) compared to the control condition and by 162.1% (*p*-value = 0.0007) compared to IL-1β treatment (Figure 3I). The increase in area staining suggests that claudins did not only localize to the junction of BLECs but displayed a more diffuse expression pattern into the BLEC cytosol. In addition, contrary to IL-1β, as observed for ZO-1 and occludin immunostaining, only TNF-α treatment appears to induce a change in BLEC morphology compared to control.

Our data indicate that TNF-α- and IL-1β-induced increases in BLEC permeability were mediated by a decrease in TJ protein expression. Only TNF-α induced a change in claudin-3 and claudin-5 localization as well as in BLEC morphology, which could explain the greater impairment of physical integrity after TNF-α treatment compared with IL-1β treatment.

### 2.4. TNF-α and IL-1β Induce Changes in the Expression of the Main Players Involved in the Transport of Aβ Peptide across the BBB

It is now well-demonstrated that Aβ peptides are exchanged at the BBB through transcellular pathways in both directions (from brain to blood, and from blood to brain) [4,6]. Because we demonstrated that CNS inflammatory cytokines affect the BBB by altering transcellular transport, we then investigated the Aβ peptide transport after TNF-α and IL-1β treatments.

We analyzed, by Western immunoblotting, the effect of TNF-α and IL-1β on the protein expression of the major receptors and transporters involved in the transport of Aβ peptides. Western immunoblotting analysis showed that the protein level of LRP1 and BCRP expression were significantly downregulated by 41.9% (*p*-value = 0.0082) and 73.4% (*p*-value < 0.0081), respectively, after TNF-α exposure (Figure 4A); and by 45.2% (*p*-value = 0.0079) and 49.0% (*p*-value < 0.0001), respectively, after IL-1β exposure, compared to control expression levels (Figure 4C).

Treatment of BLECs with TNF-α or IL-1β caused no significant change in RAGE and P-gp protein expression compared to the control (Figure 4A,C). Phosphatidylinositol binding clathrin assembly protein (PICALM) closely interacts with LRP1 and is involved in the transendothelial efflux of Aβ peptides from the brain to the blood [8]; we therefore also investigated the effects of proinflammatory cytokines on its expression. As observed in Figure 4A, TNF-α treatment did not affect PICALM protein expression, while IL-1β treatment induced an increase of PICALM protein expression (Figure 4C).

These experiments demonstrated that TNF-α and IL-1β modulate BCRP, LRP1 and PICALM expression levels in BLECs.

### 2.5. TNF-α and IL-1β Decrease Amyloid−β Peptide Efflux

As TNF-α and IL-1β modulated the protein expression levels of LRP1, BCRP and PICALM (Figure 4), we investigated the effect of these cytokines on the transport of Aβ peptide across the BBB. After 24 h of treatment with proinflammatory cytokines, the apical-to-basolateral (influx) and basolateral-to-apical (efflux) Aβ_1–40_Cy5 peptide transports across the BLECs were assessed in the presence of FITC-inulin. The Aβ_1–40_ peptide was used in all experiments, as it is the most abundant amyloid peptide found on cerebral blood vessels [28]. FITC-inulin was used to determine the nonspecific transcytosis quotient (Figure 5A). Indeed, unlike Aβ peptide, FITC-inulin is not actively cleared at the BBB and acts as a reference marker of paracellular permeability, thus as previously reported [9].

After 30 min, we detected that the basolateral-to-apical transcytosis quotient (specific efflux) of Aβ_1–40_Cy5 peptide was reduced significantly by 26.9% (*p*-value < 0.0001) after TNF-α exposure and by 25.9% (*p*-value < 0.0001) after treatment of BLECs with IL-1β (Figure 5B). The apical-to-basolateral transcytosis quotient (specific influx) of Aβ_1–40_Cy5 peptide was not modified after TNF-α and IL-1β treatment (Figure 5B). These data demonstrated that despite the changes induced by TNF-α and IL-1β on BBB permeability, they also act directly on BLECs by decreasing Aβ peptide efflux. In contrast, TNF-α and IL-1β treatments did not have a significant effect on Aβ peptide influx.

## 3. Discussion

The stability of the blood-brain barrier (BBB) is crucial for CNS homeostasis, and its dysfunction can be both the cause and consequence of neurological disorders such as Alzheimer’s disease (AD). Although several in vivo and in vitro studies have provided that TNF-α and IL-1β are able to modify BBB properties, there are still numerous controversies, especially in in vitro studies, over how these proinflammatory cytokines affect the BBB permeability. Additionally, the consequences of CNS inflammation on BBB and Aβ peptide transport need to be further investigated. Therefore, we studied how TNF-α and IL-1β separately modulate the characteristics of the BBB and contribute to the development of AD pathology by altering Aβ peptide transport. Thus, for the objective of the present study, we used the human in vitro BBB model developed in our lab (corresponding to brain-like endothelial cells; BLECs), which has the required characteristics for a BBB in vitro model such as low permeability to a BBB integrity marker, a continuous localization of tight junction proteins, functional efflux pumps, and also functional receptors and transporters to study the transport of Aβ peptides [20,21]. Using this model, we previously demonstrated that Aβ peptide efflux transport can be improved via a LRP1-mediated process after treatment of BLECs by ketone bodies [21]. We also observed that activation of the retinoid acid X receptor signaling pathway increases P-gp expression, thus resulting in a decrease in Aβ peptide entrance from the blood [29]. Thus, using this human BBB model, we incubated TNF-α and/or IL-1β in the abluminal compartment (brain side) at 10 ng·mL^−1^ for 24 h to mimic CNS inflammation. The used concentrations of 10 ng·mL^−1^ for TNF-α or IL-1β to mimic inflammation were selected based on different studies using in vitro BBB models [25,27,30,31,32,33], despite serum concentration described to be in the pg·mL^−1^ range [34]. Furthermore, the use of the same concentration between TNF-α or IL-1β in our study stems from the work of Grammas and Ovase, who demonstrated that the secretion of these two proinflammatory cytokines is equivalent (almost 30 pg·mL^−1^) in the brain microvessels of AD patients [35]. This suggests that the BBB and CNS are exposed to equivalent concentrations of TNF-α or IL-1β.

The viability and the expression of adhesion molecules, such as ICAM-1 and VCAM-1 as well as the key component of inflammation, NLRP3, were measured to assess BLEC response to inflammation [3,36]. TNF-α and IL-1β are proinflammatory factors able to induce apoptosis [37,38]. In line with the work of other teams, we found that at these concentrations and times of exposure, TNF-α and IL-1β had no deleterious effect on BLEC viability [31,39]. Upregulation of ICAM-1, VCAM-1 and NLRP3 on BLECs following TNF-α or IL-1β exposure is also in accordance with other studies [24,40,41,42]. Moreover, it was reported that these cytokines differentially regulate ICAM-1 and VCAM-1 expression on human gingival fibroblasts [43]. In accordance with this data, we also found that TNF-α upregulated ICAM-1, VCAM-1 and NLRP3 more importantly than IL-1β, suggesting that these proinflammatory cytokines regulated endothelial functions in different ways [44].

Disruption of the BBB, which we define here as modulation of BBB permeability, can be induced by several immune factors including, proinflammatory cytokines [45]. The modulation of BBB permeability in pathological conditions is mainly attributed to an increased paracellular and/or transcellular diffusion of substances due to reduced functions of tight junctions (TJ) and/or increased vesicular mechanisms, respectively [19].

Under certain inflammatory conditions, such as stroke or acute lung injury, there may be crosstalk between different pathways contributing to vascular injury [46,47]. Yet, it remains difficult to assess the contribution of paracellular and transcellular pathways in increasing BBB permeability under the effect of an inflammatory stimulus. Moreover, conflicting reports exist in several systems that complicate mechanistic interpretations of cytokine responses, in particular for TNF-α [17,25]. As previously demonstrated, studying transport at 37 °C and 4 °C provides an understanding between the two different pathways (paracellular versus transcellular) [5,48]. In line with these studies, we further aimed to clarify what type of transport across BLECs was being modulated by TNF-α and IL-1β. As expected, our findings showed that both cytokines are able to increase BLEC monolayer permeability [26,31,32,49] via both paracellular and transcellular pathways, suggesting that both ways would eventually lead to the CNS injury. Likely, as reported in vivo, these cellular events may therefore represent a general principle for how the BBB responds to injury [49]. Moreover, under these experimental conditions, the increase in permeability after TNF-α treatment appears to mainly occur by the paracellular pathway, whereas IL-1β has a stronger impact on the transcellular transport, suggesting that the modulation of the BBB may differ between mediators [40]. Furthermore, in agreement with previous studies, it is not excluded that endothelial cells respond differentially to inflammatory challenge from peripheral and CNS inflammation [50].

To provide insight on the predominating pathways, we analyzed the expression and the distribution of the TJ proteins. First, we showed that the two proinflammatory cytokines perturbed the BLECs’ barrier function, attenuating the expression of the protein levels of claudin-3, occludin and tricellulin. Moreover, we showed that IL-1β decreased the protein level of ZO-1 protein without affecting that of claudin-5, whereas TNF-α treatment did not affect the ZO-1 protein level but markedly induced the upregulation of claudin-5. These studies are in line with previous publications demonstrating reduced occludin expression after TNF-α and IL-1β treatment in brain capillary endothelial cells [31,32,33,51,52]. However, a decrease in claudin-5 protein level was observed after TNF-α and IL-1β treatment [26,32,51,52], in contrast to our study, which showed an increase and no change in claudin-5 expression under TNF-α and IL-1β treatment, respectively. The cause for this difference between our study and those of other teams may reflect cell-type-specific variation and the compartment where the proinflammatory cytokines are placed to mimic inflammation (luminal/apical or abluminal/basolateral), as well as differences in the length and dose of cytokine treatment [27]. Furthermore, corroborating our findings where claudin-5 expression is increased by TNF-α, it is reported that its expression can indeed increase due to acute inflammatory stimuli paralleled by decreased expression of other junctional proteins, likely in an attempt by the BBB to preserve its integrity when facing an inflammatory challenge, as claudin-5 is probably the most relevant tight junction protein in preserving BBB homeostasis [30,53,54,55]. However, our study demonstrated no change in claudin-5 expression under the effect of IL-1β treatment, again suggesting that the activation of IL-1β signaling pathways is different from that for TNF-α within the BBB. Furthermore, proinflammatory cytokine-induced modulation of BLEC permeability may be related to the localization of TJ proteins [25,26,27]. In concordance with the functional measurements of BBB permeability, only TNF-α treatment induces a change in BLEC morphology and an increase of claudin-3 and claudin-5 cytoplasmic staining, which correlates with different studies that have shown a delocalization of these claudins, leading to TJ destabilization and permeability increase [25,27,31,56]. In contrast, IL-1β did not cause changes in BLEC morphology and TJ protein localization, confirming that the alteration of BBB permeability is mainly due to a transcellular, rather than paracellular, pathway. Unfortunately, our study does not allow us to discern the precise nature of the cellular or molecular consequences of increased transcellular transport of IL-1β at the BBB. Further investigations are needed to define this pathway and the degree of specificity of inflammatory stimuli on BLECs.

In addition to causing physical changes at the BBB, pro-inflammatory cytokines can also alter functional receptors/transporters of the BBB [57]. Yet, Aβ peptide transport at the BBB is dependent on the activities of the RAGE and LRP1 receptors, the P-gp and BCRP transporters and the adaptor protein involved in clathrin-mediated endocytosis, PICALM, all of which regulate the blood-to-brain (influx) or brain-to-blood (efflux) direction. Their expressions are often observed as being deregulated in AD [6,8,10,12,58] and during inflammation [18,59], which is considered a risk factor for this pathology [60].

Here, we demonstrated that TNF-α and IL-1β, added in the basolateral compartment (brain side), decreased LRP1 and BCRP protein levels in BLECs without affecting those of RAGE and P-gp, On the other hand, the two proinflammatory cytokines differentially modulated the protein level of PICALM.

Several studies have reported a downregulation of LRP1 in the brain microvasculature in patients and mouse models of AD [6,10] as well as with peripheral inflammation [61]. In support of our results, Kitazawa and colleagues demonstrated that TNF-α and IL-1β mediated the downregulation of LRP1 in human primary microvascular endothelial cells (MVECs), without, however, showing the impact of these cytokines on Aβ peptide transport [62]. Recently, Heng-WeiHsu and colleagues confirmed these results, suggesting that this reduction was in part mediated by microRNA-205-5p, -200b-3p and -200c-3p [63]. In addition, even in streptozotocin-induced diabetes, another pathology associated with neuroinflammation, a decrease in LRP1 expression at the level of the BBB was observed [64]. Our results and others thus suggest that neuroinflammation induced by the proinflammatory cytokines promote LRP1 downregulation at the BBB.

Unlike LRP1, RAGE expression levels are upregulated in AD and in chronic inflammation [4,65]. The present study found that the protein expression of RAGE was not affected by the added of TNF-α or IL-1β. In vitro experiments in which LPS was placed in the apical (blood) compartment confirmed these results [66]. These studies suggested that signals that can bind directly to RAGE can induce its gene expression and have the potential to increase the influx of Aβ [67]. We have previously demonstrated that RAGE is exclusively expressed in the apical (blood) side of the endothelial cells [5,20], the opposite side to where proinflammatory cytokines are placed, i.e., the basolateral (brain) side. Under these conditions, it would appear that the BBB has a polarized inflammatory response, which argues that “CNS” cytokines may have a different impact than “peripheral” cytokines on endothelial function and BBB modulation [50].

Similar to RAGE, both P-gp and BCRP are mainly expressed on the blood side of the BBB [1]. Their regulation is complex and their expression has been shown to increase or decrease in brain endothelial cells during inflammation [57]. Our study is consistent with previous studies that have shown a decrease in BCRP expression within the human hCMEC/D3 cell line and primary culture of porcine brain capillaries under TNF-α or IL-1β treatment [18,68]. In accordance with our results, no variations in P-gp expression were found following a prolonged exposure to TNF-α in two other human cell lines (iHBMEC and phBMEC) [69]. In the same way, no variation in P-gp expression was found following exposure to IL-1β in the human hCMEC/D3 [18]. By contrast to our results, an upregulation and downregulation of P-gp expression was observed after exposure of the human hCMEC/D3 cell line to TNF-α for 72 h treatment [18] and after an exposure of 24 h to TNF-α in porcine brain endothelial cells (PBECs) [68], respectively. Thus, this suggests that the contradictory observations would be dependent on different parameters such as the model used and the duration of the proinflammatory treatment. In addition, species differences and complex compensatory regulatory mechanisms cannot be also excluded [27,70,71].

Another interesting finding in the present study is that the two proinflammatory cytokines differentially modulate PICALM protein expression. PICALM has a key role in clathrin-dependent endocytosis processes and is involved in the clearance of Aβ peptides through the BBB, including LRP1-mediated transcytosis of Aβ peptides [8,9]. To our knowledge, no studies have yet reported on the effects of cytokines on PICALM expression. In addition, consistent with the function of PICALM and our results, we demonstrated that TNF-α treatment does not affect PICALM protein expression in contrast to IL-1β, which induces an increase in its expression, correlating with a greater increase in transcellular permeability observed under IL-1β than under TNF-α. The study to assess the effect of proinflammatory cytokines on PICALM activation and the consequences on BBB modulation would need to be further investigated.

We next hypothesized that TNF-α and (or) IL-1β were able to modulate the transport of Aβ peptide across the BBB. For the first time, we demonstrated that proinflammatory cytokines TNF-α and IL-1β act on BLECs by decreasing the specific efflux of Aβ_1–40_Cy5. These findings agree with similar results obtained in transgenic mice where inflammation induced by injecting LPS prompted Aβ peptide deposition by decreased brain efflux of Aβ [61]. Notably, in that study, LPS did not appear to act directly on Aβ peptide transport at the BBB. It cannot be excluded that due to the positive effect of LPS on NLRP3-mediated IL-1β production, the effect observed by LPS is partly a consequence from proinflammatory cytokines produced locally in the CNS [72]. In support of this previous hypothesis, inhibition of NLRP3 by a specific inhibitor, MCC950, improved amyloid plaque pathology and cognitive function in AD mouse models [73,74]. Furthermore, a recent study has reported that pharmacological inhibition of TNF-α in 5XFAD/Tg197 mice had reduced amyloid deposition, suggesting a direct causal linkage between the Aβ transport defect and inflammation [75]. Taken together, in line with our results, these reports suggest the potential involvement of proinflammatory cytokines in decreasing Aβ peptide clearance across the BBB.

On the other hand, no change in the specific influx of Aβ_1–40_Cy5 transport was observed under our experimental conditions, consistent with the absence of TNF-α- or IL-1β-mediated modulation of RAGE expression [4,61].

As already mentioned, LRP1 and BCRP have an important role in Aβ peptide clearance through the BBB [6,76,77,78]. As TNF-α and IL-1β downregulated the protein expression levels of LRP1 and BCRP, we assumed that they are two of the main players involved in this process. Indeed, we recently showed that ketone bodies enhanced the expression level of LRP1, which promote the efflux of Aβ_1–40_Cy5 transport in our human in vitro BBB model [21]. In these conditions, only when LRP1 is upregulated is it possible to inhibit this receptor in order to restore Aβ peptide transport [21]. On the other hand, activity levels of P-gp and BCRP in this BBB model have been extensively characterized in several of our previous works [20,21,29]; therefore, we do not exclude that Aβ peptide efflux transport is also impacted by the decrease in BCRP expression, the latter representing a compensatory player of P-gp [79]. In addition, although BCRP has been associated with a restriction of Aβ peptide influx at the BBB [7,80], our study shows that the decrease in BCRP does not impact the influx of the Aβ peptide, which highlights the presence of compensatory mechanisms of ABC transporters to preserve the BBB’s function. However, although LRP1 and BCRP appear to be naturally involved in the decrease in Aβ peptide efflux, we cannot exclude the possibility that other receptors and transporters such as ABCA1 or ABCA7 [81] may be involved in this process [29,82]. Future studies on the effects of cytokines on these transporters are in progress.

In summary, TNF-α- or IL-1β-mediated inflammatory stress are able to alter BBB permeability through paracellular and transcellular pathways. The increase in endothelial barrier permeability after TNF-α treatment seems to occur mainly through the paracellular pathway, whereas IL-1β has a stronger impact on transcellular transport, suggesting that BBB modulation may differ between mediators. Further, TNF-α and IL-1β act directly on BLECs by decreasing the efflux of Aβ peptide, probably due to decreased expression of the LRP1 and BCRP, which are responsible for Aβ peptide clearance. These results provide mechanisms by which brain inflammation may modulate BBB permeability and promote Aβ peptide accumulation and progression of AD.

## 4. Materials and Methods

Endothelial cell medium (ECM; Sciencell, Carlsbad, CA, USA) was supplemented with 50 μg·mL^−1^ gentamicin (Biochrom GmbH, Berlin, Germany), 5% endothelial cell growth factor (#1052; from Sciencell, Carlsbad, CA, USA) and 5% lab-tested fetal calf serum (FCS, GIBCO, Life Technology SAS, Saint Aubin, France), corresponding to ECM-5. ECM-5 was prepared and stored at 4 °C for a maximum of one week. Tumor necrosis factor-α (TNF-α, #SRP3177) and interleukin-1β (IL-1β, #SRP3083) were purchased from Sigma-Aldrich (Lyon, France). TNF-α and IL-1β were dissolved in ECM-5 (according to the manufacturer’s instructions) and stored at –80 °C until use. Human serum albumin (HSA, #A8763), lucifer yellow (LY, Lucifer Yellow CH dilithium salt, #L0259), MTT [3-(4,5-dimethylthiazol-2-yl)-2,5-diphenyltetrazolium bromide] (#M-2128) and FITC-inulin (#F3272) were also purchased from Sigma-Aldrich (Lyon, France). Fluorescent human amyloid-β peptide (1–40)-Cy5, labeled (Aβ_1–40_Cy5; #FC5-018-01), was purchased from Phoenix Pharmaceuticals (Strasbourg, France). Before use, Aβ_1–40_Cy5 was resuspended in DMSO (Sigma-Aldrich, Lyon, France) and diluted in a physiological buffer, Ringer-Hepes HSA 0.1% (RH; 150 mM NaCl, 5.2 mM KCl, 2.2 mM CaCl_2_, 0.2 mM MgCl_2_ 6H_2_O, 6 mM NaHCO_3_, 5 mM HEPES, 2.8 mM glucose; pH: 7.4) to obtain a solution at 1 μM. Then, this solution was diluted to obtain a working solution at 10 nM in RH-HSA 0.1% buffer.

### 4.1. The Human In Vitro BBB Model

As described previously [21], the human in vitro blood-brain barrier (BBB) model based on the acquisition of the human brain-like endothelial cell (BLEC) phenotype was used for this study [20]. This BBB model corresponds to a coculture of primary human endothelial cells (ECs) derived from CD34^+^ cord blood hematopoietic stem cells and bovine brain pericytes, isolated as described by Vandenhaute and colleagues [83]. Informed consent was obtained from all infants’ parents for the umbilical cord blood collection and use. Procedures for isolation of cells from umbilical cord were approved by the French Ministry of Higher Education and Research (CODECOH DC2011-1321, approved 31 January 2013) and by the local investigational review board (Béthune Maternity Hospital, Beuvry, France). Briefly, 8 × 10^4^ ECs at passage 6 were seeded onto Matrigel-coated (BD Biosciences, San Jose, CA, USA) insert (pore size 0.4 μm, Costar Transwell inserts, Corning Inc., Corning, New York, NY, USA. Endothelial cells were cocultured with bovine brain pericytes at passage 10 in a 12-well plate gelatin-coated with ECM-5. Every two days, ECM-5 was changed until experiment. However, a minimum of 5 days of culture were required for the ECs to develop the BLEC phenotype and a maximum of 21 days of BLEC phenotype stability has been determined. Once BLEC phenotype was acquired, these cells delimited two compartments, one mimicking the blood side (corresponding to the apical compartment) and the other the brain side (corresponding to the basolateral side). A “quality control” of the human in vitro BBB model was also performed by validating the absence of the mycoplasma using MycoAlert^TM^ Mycoplasma Detection kit (Lonza, Rockland, ME, USA).

### 4.2. Treatment with Proinflammatory Cytokines: TNF-α or IL-1β

After 6 days of coculture, TNF-α or IL-1β were added to the basolateral compartment (brain side) at 10 ng·mL^−1^ for 24 h, at 37 °C and 5% CO_2_, to mimic CNS inflammation condition. Each condition was studied in triplicate.

### 4.3. Cell Viability Assay

The standard MTT [3-(4,5-dimethylthiazol-2-yl)-2,5-diphenyltetrazolium bromide] assay (AR1156, Boster Biological Technology, Pleasanton, CA, USA) was performed to determine BLEC viability. Briefly, after coculture treatments, culture medium was removed and replaced with worm Ringer-Hepes (RH; 150 mM NaCl, 5.2 mM KCl, 2.2 mM CaCl_2_, 0.2 mM MgCl_2–_6H_2_O, 6 mM NaHCO_3_, 5 mM HEPES, 2.8 mM glucose; pH: 7.4) in basolateral compartment and with MTT solution (0.5 mg·mL^−1^) in RH at in the opposite compartment for 3 h at 37 °C and 5% CO_2_. Then, buffer was removed and 300 µL/filter of DMSO was added to the apical compartment to dissolve the purple formazan crystals formed in the BLECs from MTT. Aliquot was sampled in 96-well plate and the optical density (OD) was measured with a microplate reader (Synergy H1, BioTek. Colmar, France) at the wavelength of 570 and 630 nm. The percentage of relative cell viability compared to the control condition was subsequently determined using the following formula: % relative cell viability = ((OD570 − OD630) _condition_)/((OD570 – OD630)_control_) × 100. 

### 4.4. Evaluation of BLEC Monolayer Permeability

The BLEC monolayer integrity was evaluated after each treatment using the method described by Dehouck and colleagues [84]. The endothelial permeability coefficient (Pe) was evaluated regarding the apical-to-basolateral transport of the LY integrity marker used at a final concentration of 50 µM in RH buffer at 37 °C. Briefly, the inserts containing BLECs were transferred to a new 12-well plate containing warm RH buffer (corresponding to the basolateral compartment), and then a volume of 500 µL of warm RH with 50 µM of LY buffer was added to the apical compartment. The inserts containing the BLECs and RH buffer with 50 µM of LY were then transferred to a new well containing warm RH every 20 min for 1 hour. The aliquot of each compartment of the BBB model was collected and the LY fluorescence was quantified using a microplate reader (Synergy H1, Biotek, Colmar, France) at the 432/538 nm (excitation/emission) wavelength. For each condition, coculturing was performed in triplicate. To determine the PeLY values for BLECs, the slope of the clearance curves (PS = permeability × surface area product; in µL·min^−1^) corresponding to the volume cleared from the apical to the basolateral compartment (in µL) for 60 min was calculated for the filters with only Matrigel coating and for those with Matrigel coating and BLECs, denoted PSf and PSt, respectively. From these values, the PS values for BLECs (PSe) were determined according to the formula “1/PSe = 1/PSt + 1/PSf”. Then, the PeLY values for BLECs (in cm·min^−1^) were obtained by the calculation “PeLY = PSe/S”, with PSe in cm^3^·min^−1^ and S being the filter surface area (= 1.12 cm^2^). The same experiment was also performed at 4 °C to block the transcellular permeability of LY and to identify only the paracellular permeability of LY. Following these experiments, the transcellular permeability can be deduced, which corresponds to the PeLY values at 37 °C subtracted by those of PeLY at 4 °C. The mass balance (in %) was determined from the amount of LY in the apical and basolateral compartment at the end of the experiment divided by the total amount initially added in the apical compartment at time zero. To be taken into consideration for the Pe determination, the mass balance should be between 80% and 120%. 

### 4.5. Immunostaining

After 24 h of treatment with or without proinflammatory cytokines, BLECs were fixed and permeabilized, as described in Table 1. Then, a blocking step was performed with Sea Block Blocking Buffer (#37527 from Thermofisher, Rockford, IL, USA) for 30 min. BLECs were incubated for 1 h with the primary antibody in phosphate buffered saline, calcium-magnesium-free (PBS-CMF; 0.2 g·L^−1^ KH_2_PO_4_, 8.0 g·L^−1^ NaCl, 2.87 g·L^−1^ Na_2_HPO_4–_12H_2_O, and 0.2 g·L^−1^ KCl) with 2% (*v/v*) normal goat serum (NGS) at room temperature (RT) against targeted proteins, as described in Table 1. 

After three washes with PBS-CMF supplemented with 2% NGS, BLECs were incubated with secondary polyclonal antibody (goat anti-rabbit Alexa 568, A11036, Molecular Probes, Eugene, OR, USA) diluted 1:500 in PBS-CMF supplemented with 2% NGS in the dark for 30 min at RT. Then, ProLong™ Diamond Antifade Mountant with DAPI (#P36962, Invitrogen) was used for nuclei staining and mounting. The stained slides were conserved in the fridge. The staining images were taken using a Leica microscope (DMRD; Leica Microsystems, Wetzlar, Germany) and processed using ImageJ software.

Claudin-3 and claudin-5 area fraction staining were subsequently quantified within the BLECs using ImageJ software. Staining images from the same experiment were taken with the same threshold according to the staining, and could thus be compared with each other. Due to the absence of BLEC delineation, the total fraction area staining for each image was thus quantified and normalized by the number of nuclei. Studies were performed with at least three independent experiments and at least three randomized staining images per experiment containing around 50 BLECs each.

### 4.6. RT-qPCR

BLECs were washed twice with cold RH buffer, then lysed using RLT lysis buffer (Qiagen, Les Ulis, France). The cell lysates were extracted according to the manufacturer’s instructions “Qiagen RNeasy Mini kit”. Briefly, a volume of cell lysates was mixed with a volume of 70% (*v/v*) ethanol and placed in a column (Rneasy Mini kit, Qiagen, Les Ulis, France). A first centrifugation was performed to adsorb the RNAs to the column. A series of two rinses with buffer RW1 and a centrifugation (10.000 rpm, 30 s) were performed. Samples were treated with DNase in its RDD buffer for 20 min. Then, another succession of rinses (one with buffer RW1 and two with buffer RPE) and centrifugation (10.000 rpm, 30 s) were performed. Finally, water (RNAse Free) was added to the column to elute the RNAs by centrifugation (13.000 rpm, 2 min at 20 °C). The amount of extracted mRNAs was assayed by spectrophotometry (Synergy H1, Biotek, Colmar, France). Absorbances at 280 nm (proteins) and 260 nm (nucleic acids) were performed. The mRNA concentration was determined from the Beer-Lambert law and the mRNA purity of the sample was assessed with the Absorbance 260 nm/Absorbance 280 nm ratio. This ratio must be close to 2.1 to consider the sample to be with high purity. The mRNA samples were then stored at −80 °C. 

Reverse transcription (RT) was performed on the samples containing the extracted mRNAs to obtain cDNAs. A total of 250 ng of mRNA was reverse-transcribed using iScriptTM reverse Transcription Supermix 5X (Biorad, Munich, Germany), containing reverse transcriptase, RNAse inhibitors, RNAse H, dNTPs (nucleoside triphosphates), oligonucleotides (dTs) (deoxy-thymine repeats) and random primers (hexamers of dNTPs randomly binding to mRNAs) and MgCl_2_). To each microtube, a volume of water (RNAse free) was added to obtain a final volume of 20 µL, then a reverse transcription cycle was performed in the thermal cycler (MJ Research): 5 min at 25 °C (hybridization), 30 min at 42 °C (reverse transcription) and 5 min at 85 °C (reverse transcription inactivation). The cDNA samples were then stored at −20 °C. For each reverse transcription and for each condition, an internal control was performed by adding iScriptTM No-RT control Supermix 5X (Biorad, Munich, Germany), a solution without the reverse transcriptase enzyme (NRT), instead of iScriptTM reverse Transcription Supermix 5X. 

The qPCR was performed on a 96-well plate where each sample was deposited in triplicate. To perform qPCR, human-specific primer pairs were drawn for each gene of interest (Table 2).

An efficiency test and a melting curve were performed for each pair of primers used using the CFX Manager software (Biorad, Munich, Germany) allowing for the specificity and efficiency of the primers to be checked. A qPCR of the cDNAs obtained by RT-PCR was performed using the specific primers and a SsoFast^TM^ Evagreen SuperMix (Biorad, Munich, Germany) containing DNA polymerase, its cofactors, dNTPs and Evagreen^®^ (a fluorochrome that binds to DNA during elongation). The amplification was carried out for 40 cycles with a hybridization temperature of 60 °C. The reaction was performed in a CFX96 Real-Time System thermal cycler (Biorad, Munich, Germany). Gene expression levels were assessed using the ΔΔCt method and normalized to a reference gene, cyclophilin A. The data were then plotted as a percentage against the control condition. Two controls were performed per primer pair: an NRT and a primer hybridization control (NTC), where the sample containing the cDNAs is replaced with water. 

### 4.7. Western Immunobloting

After 24 h of treatments with or without proinflammatory cytokines, BLECs were washed twice with cold RH buffer and scraped with 70 μL of RIPA 1X lysis buffer (10 mM Tris-Cl (pH 8.0), 1 mM EDTA, 0.5 mM EGTA, 1% Triton X-100, 0.1% sodium deoxycholate, 0.1% SDS, 140 mM NaCl) (Merck Millipore, Burlington, MA, USA) or UT4 lysis buffer (7M Urea, 2M Thiourea, 4% CHAPS), both supplemented with a cocktail of phosphatase and protease inhibitors (1% protease inhibitors cocktail #P8340; 1% phosphatase inhibitors cocktail 1 P2850; 1% phosphatase inhibitors cocktail 2 #P5726; 1% protease inhibitors cocktail #P8340) purchased from Sigma-Aldrich (Lyon, France). Then, a centrifugation at 10,000 rpm for 10 min at 4 °C was achieved to save the supernatant of the lysate. This was later sonicated two times for 5 s on ice. The concentration of the protein lysate was quantified using the Bradford method (#5000006; rad, Munich, Germany). The protein samples were then stored at −20 °C. 

20 μg of total protein was mixed with Laemmli 4X (#1610747; Biorad, Munich, Germany) supplemented with β-mercaptoethanol (Sigma-Aldrich, Germany). Depending on the protein target, details of the denaturation and reduction steps of the protein extract are described in Table 3. Then, the protein mixes were electrophoresed using Criterion™ TGX™ (Tris-Glycine eXtended) precast gels (Biorad, Munich, Germany) and subsequently electrotransferred to nitro-cellulose membranes (GE Healthcare, Munich, Germany). Once the previous step was completed, Tris-buffered saline 0.1% Tween 20 (TBST; 20 mM Tris-HCl, pH 8.0, 500 mM NaCl, 0.1% Tween 20) supplemented with 5% skimmed milk was added on membrane to block nonspecific binding sites for 1 h under agitation at RT. Membranes were incubated in the appropriate primary antibodies (as described in Table 3) and incubated at 4 °C overnight, except β-actin (20 min at RT). 

The following morning, the membranes were rinsed with TBST three times for 5 min each under agitation, and then incubated with the horseradish peroxidase-conjugated secondary antibody anti-rabbit (1:8000, Dako/Agilent Technologies, Inc., Santa Clara, CA, USA) or secondary antibody anti-mouse (1:5000, Dako/Agilent Technologies, Inc., Santa Clara, CA, USA) for 1 h at RT. Subsequently, the membranes were rinsed with TBST three times for 5 min each and bands of immunoreactive protein were visualized after membrane incubation with enhanced chemiluminescence (GE Healthcare, Munich, Germany) reagent during 5 min and revealed by the Western immunoblotting Imaging system Azure c600 (Azure Biosystems, Dublin, Ireland). Quantification of the relative densities of bands was performed with TotalLab TL 100 1D Gel Analysis software (Nonlinear Dynamics, Newcastle, UK). The protein expression normalization was conducted with anti-β-actin. Studies were performed with at least three independent experiments, with at least two Western immunoblottings per experiment.

### 4.8. Amyloid-β (1–40) Peptide Transport STUDIES

After proinflammatory cytokine treatments, the influx (apical-to-basolateral) and the efflux (basolateral-to-apical) transport of Aβ_1–40_ peptides across BLECs were investigated as previously described [21]. Briefly, for influx studies, inserts with BLECs were transferred to new 12-well plates containing 1.5 mL of RH–HSA 0.1% (the receiver solution) per well. In the insert, 0.5 mL of RH-HSA 0.1% supplemented with 10 nM Aβ_1–40_Cy5 peptide were added (the donor solution). For efflux studies, inserts with BLECs were transferred to new 12-well plates, this time containing 1.5 mL of RH-has 0.1% with 10 nM fluorescent Aβ_1–40_Cy5 (the donor solution) peptide, then 0.5 mL of RH-HSA 0.1% (the receiver solution) was added in the insert. In parallel, 1 µM FITC-inulin was added in Aβ_1–40_Cy5 peptide buffer for influx and efflux experiments, thus representing the nonspecific transport part. The absence of modification of Aβ_1–40_Cy5 peptide transport in presence of FITC-inulin was validated, and likewise in reverse. The BBB integrity was determined by measuring the endothelial permeability of a radioactive integrity marker, ^14^C-sucrose, using the same method of calculation as described in Section 4.4. All transport studies were performed in triplicate on a rocking platform at 37 °C for 30 min with slight agitation. At the end of the experiment, aliquots of the receiver and donor solutions were collected, and the fluorescence compounds (Aβ_1–40_Cy5 peptide and FITC-inulin) were quantified with a spectrofluorimeter (Synergy H1, Biotek, Colmar, France). The radioactivity of ^14^C-saccharose was measured using a scintillation counter (Hidex 300 SL, Hidex, Turku, Finland). The apparent permeability coefficients (cm.sec^−1^) of Aβ_1–40_Cy5 peptide and FITC-inulin were calculated as previously described [21]. Once influx and efflux Papp were determined for each molecule, the specific transport of Aβ_1–40_Cy5 across the BLECs in influx and efflux directions were identified by calculating the transcytosis quotient of Aβ_1–40_Cy5 using the following formulas: “influx transcytosis quotient of Aβ_1–40_Cy5 = (influx Aβ_1–40_Cy5 Papp)/(influx FITC-inulin Papp)” or “efflux transcytosis quotient of Aβ_1–40_Cy5 = (efflux Aβ_1–40_Cy5 Papp)/(efflux FITC-inulin Papp)”. For each filter, the mass balance was determined as described in Section 4.4. Experiments were performed on at least three independent experiments in triplicate for each condition. 

### 4.9. Statistical Analysis

The results of experiments are presented as the mean ± standard error of the mean (SEM). All statistical analyses were performed using the GraphPad Prism 5.01 statistical software package (GraphPad Software, Inc., San Diego, CA, USA). Data obtained from RT-qPCR study were evaluated using unpaired *t*-test, while for the rest, Mann-Whitney *t*-test was used. The threshold for statistical significance was set to * *p* < 0.05, ** *p* < 0.01 or *** *p* < 0.001. 

## Figures and Tables

**Figure 1 ijms-23-10235-f001:**
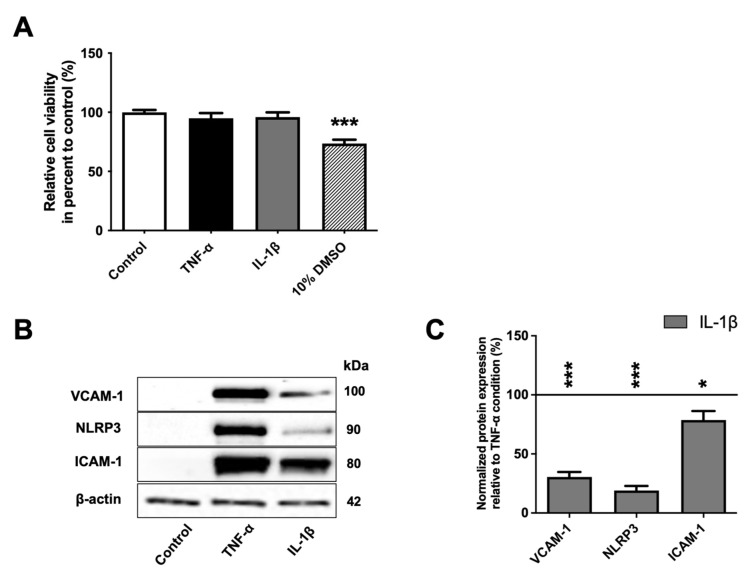
BLECs’ response to TNF-α and IL-1β treatment. (**A**) After 24 h, the effects of 10 ng·mL^−1^ of TNF-α or IL-1β on cell viability were performed using MTT assay. A 10% DMSO treatment was used as positive control of cell death. (**B**) The proinflammatory markers (VCAM-1, NLRP3 and ICAM-1) after TNF-α or IL-1β treatments were observed using Western immunoblotting. The immunoblotting images are representative of at least three independent experiments. (**C**) The protein levels of the BLEC proinflammatory markers were also quantified compared among proinflammatory treatments. The black line corresponds to the TNF-α condition value set at 100%. For A and C graphs, each bar represents the mean ± SEM and are representative of at least three independent experiments performed in triplicate. The Mann-Whitney *t*-test was used for the interpretation of statistical data with a threshold of statistical significance compared to the control condition or among proinflammatory cytokines set at * *p* < 0.05 or *** *p* < 0.001. VCAM-1: vascular cell adhesion protein-1; NLRP3: « NOD-like receptor family, pyrin domain containing 3; ICAM-1: intercellular adhesion molecule-1.

**Figure 2 ijms-23-10235-f002:**
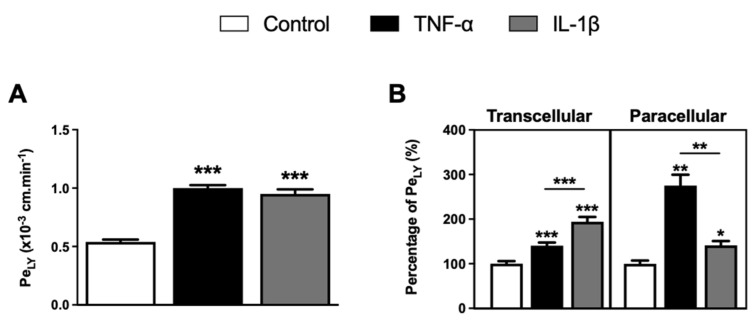
Effects of TNF-α or IL-1β treatments on BLEC monolayer permeability. (**A**) Following proinflammatory cytokine treatments, BLEC monolayer integrity was determined by measuring the endothelial lucifer yellow permeability (Pe_LY_). (**B**) The paracellular permeability of LY through BLEC monolayer was also determined by measuring the Pe_LY_ at 4 °C, which corresponds to a condition-inhibiting transcellular pathway. The transcellular permeability of LY through BLEC monolayer was deducted by subtracting the total permeability of LY from its paracellular permeability. The percentage of Pe_LY_ values was established in relation to their control condition. The control values for total, paracellular and transcellular Pe_LY_ are equal to 0.54 ± 0.02 × 10^−3^, 0.18 ± 0.01 × 10^−3^ and 0.36 ± 0.02 × 10^−3^ cm·min^−1^, respectively. Each bar represents the mean ± SEM and is representative of at least three independent experiments performed in triplicate. The threshold for statistical significance compared to the control condition or among proinflammatory cytokines was set to * *p* < 0.05; ** *p* < 0.01; *** *p* < 0.001.

**Figure 3 ijms-23-10235-f003:**
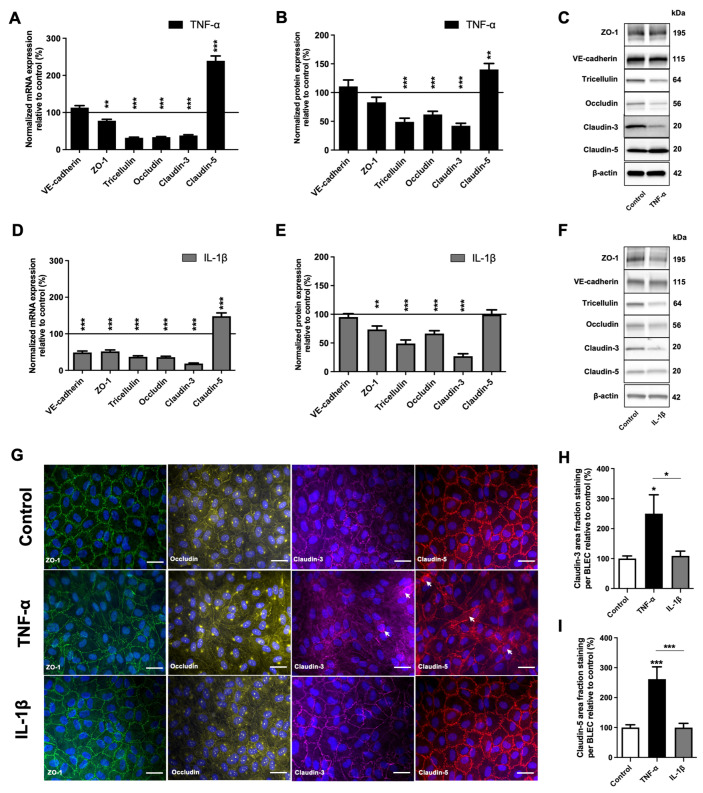
Effects of TNF-α or IL-1β treatments on the expression and the localization of junctional protein in BLECs. (**A**–**C**) After 24 h of 10 ng·mL^−1^ TNF-α treatment, mRNA (**A**) and protein level (**B**) of the major protein actors involved in the BBB physical integrity (tight junction protein: ZO-1, tricellulin, occludin, claudin-3, claudin-5; adherent junction protein: VE-cadherin) were determined by RT-qPCR and Western immunoblotting, respectively, compared to the control condition. (**C**) The immunoblotting images are representative of at least three independent experiments. (**D**–**F**) As in the above study, after a treatment of 10 ng·mL^−1^ IL-1β over 24 h, the mRNA (**D**) and protein levels (**E**,**F**) of the same targets were also quantified compared to the control condition. For each of the above graphs (**A**,**B**,**D**,**E**), the black lines correspond to the control condition value set at 100%. (**F**) The immunoblotting images are representative of at least three independent experiments. (**G**) Staining of ZO-1 (green), occludin (yellow), claudin-3 (magenta) and claudin-5 (red) were obtained using immunofluorescence. Each white arrow indicates the presence of stain cytoplasmic dot. Nuclei appear in blue. Scale bar: 50 μm. (**H**) Claudin-3 and (**I**) claudin-5 area fraction stainings were quantified in BLECs after exposure to TNF-α or IL-1β and were compared to respective control conditions. For all the above graphs, each bar is representative of at least three independent experiments performed in triplicate. Each bar represents the mean ± SEM relative to the control. The unpaired t test was used for mRNA study, while Mann-Whitney *t*-test was used for protein study. The threshold for statistical significance compared to the control condition was set to * *p* < 0.05; ** *p* < 0.01; *** *p* < 0.001. VE-cadherin: vascular endothelial-cadherin; ZO-1: zonula occludens-1.

**Figure 4 ijms-23-10235-f004:**
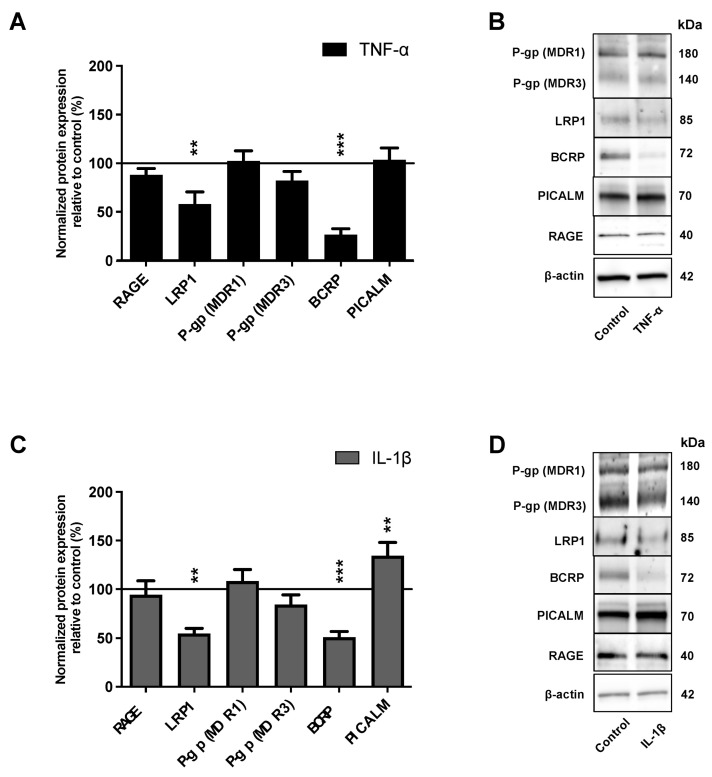
Impacts of TNF-α or IL-1β treatments on the protein level of the actors involved in the Aβ peptide transport across BBB. After 24 h of 10 ng·mL^−1^ (**A**) TNF-α or (**C**) IL-1β treatments, the protein levels of the major actors involved in Aβ peptide transport at the BBB level (RAGE, LRP1, P-gp, BCRP and PICALM) were determined by Western immunoblotting. The black lines correspond to the control condition values (100%). Each bar represents the mean ± SEM relative to the control conditions. Each bar represents the mean ± SEM relative to the control conditions and is representative of at least three independent experiments performed in triplicate. The Mann-Whitney *t*-test was used for the interpretation of statistical data with a threshold of statistical significance compared to the control condition set at ** *p* < 0.01; *** *p* < 0.001. (**B**,**D**) The immunoblotting images are representative of at least three independent experiments. RAGE: receptor for advanced glycation endproducts; LRP1: Low-density lipoprotein receptor-related protein 1; P-gp: permeability-glycoprotein; BCRP: breast cancer resistance protein; PICALM: phosphatidylinositol binding clathrin assembly protein.

**Figure 5 ijms-23-10235-f005:**
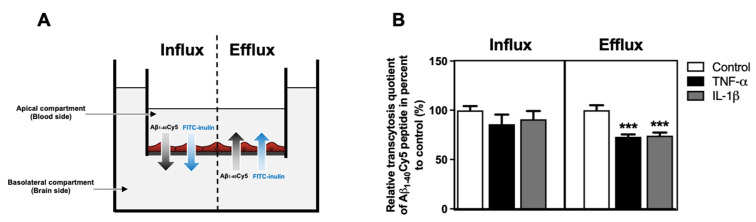
Impacts of TNF-α or IL-1β treatments on Aβ_1–40_ peptide transport across BLECs. (**A**) Schematic representation of the influx (apical-to-basolateral compartment) and efflux (basolateral-to-apical compartment) Aβ_1–40_Cy5 peptide/FITC-inulin transport experiment across the BLECs for 30 min. (**B**) After 24 h of TNF-α or IL-1β treatments, Aβ_1–40_Cy5 peptide and FITC-inulin were added in apical or basolateral compartment of the in vitro human BBB model, respectively, for influx or efflux transport, and permeability was then assessed. Apparent permeability coefficient values (Papp) were measured for Aβ_1–40_Cy5 peptide and FITC-inulin transport, with Papp_control_ equal to 5.14 ± 0.32 and 1.67 ± 0.06 × 10^−6^ cm·min^−1^ for influx transport and 4.47 ± 0.16 and 1.41 ± 0.03 × 10^−6^ cm·min^−1^ for efflux transport, respectively. Afterwards, permeability values were used to determine a relative transcytosis quotient of Aβ_1–40_Cy5 peptide through BLECs compared to the control condition in the influx and efflux directions, equal to 3.10 ± 0.25 and 3.22 ± 0.13, respectively. Each bar represents the mean ± SEM relative to the control conditions and is representative of at least three independent experiments performed in triplicate. The Mann-Whitney *t*-test was used for the interpretation of statistical data with a threshold of statistical significance compared to the control condition set at *** *p* < 0.001.

**Table 1 ijms-23-10235-t001:** Antibodies used for the immunostaining experiment. ZO-1: zonula occludens-1.

Protein Target	Antibody Reference	Fixation/Permeabilization	Antibody Dilution
Claudin-3	34–1700 (Invitrogen)	Ice-methanol 1′	1:40
Claudin-5	34–1600 (Invitrogen)	Ice-methanol 1′	1:100
Occludin	71–1500 (Invitrogen)	Ice-methanol 1′	1:50
ZO-1	61–7300 (Invitrogen)	Paraformaldehyde 1% 10′; 3 PBS-CMF rinses of 5′; Triton X100 0.1% 10′	1:200

**Table 2 ijms-23-10235-t002:** Nucleotide sequences of primer pairs used for qPCR. F: forward (forward primer); R: reverse (reverse primer). VE-cadherin: vascular endothelial-cadherin; ZO-1: zonula occludens-1.

Target	Sequence (5′−3′)	Efficiency (%)
Cyclophilin A	F: CTGAGGACTGGAGAGAAAGGAT R: GAAGTCACCACCCTGACACATA	106.3
Claudin-3	F: AGGCGTGCTGTTCCTTCTC R: TTGTAGAAGTCCCGGATAATGG	97.9
Claudin-5	F: GAGGCGTGCTCTACCTGTTTT R: CACAGACGGGTCGTAAAACTC	108.7
Occludin	F: GAGGCTATGGAACTTCCCTTTT R: TAGCTACCAAAGCCACTTCCTC	95.6
Tricellulin	F: GTACTCGTGGTTGCTGGATTAG R: GCCACCAATTAGAGTCCAGAAG	112.3
VE-cadherin	F: GATCTCCGCAATAGACAAGGAC R: TCCGTGAGGGTAAAGTTGTTCT	107.7
ZO-1	F: CTCATTTTC AGAGTGGGGAAAC R: GGTCATTTTCCTGTAGCTGTCC	86.1

**Table 3 ijms-23-10235-t003:** Antibodies used for Western immunoblotting experiment. BCRP: breast cancer resistance protein; ICAM-1: intercellular adhesion molecule-1; LRP1: Low-density lipoprotein receptor-related protein 1; NLRP3: « NOD-like receptor family, pyrin domain containing 3; P-gp: permeability-glycoprotein; PICALM: phosphatidylinositol binding clathrin assembly protein; RAGE: receptor for advanced glycation endproducts; VCAM-1: vascular cell adhesion protein-1; VE-cadherin: vascular endothelial-cadherin; ZO-1: zonula occludens-1.

Protein Target	Antibody Reference	Lysis Buffer	Special Condition	Antibody Dilution	Size (kDa)
BCRP	Ab3380 (Abcam)	RIPA	Without heat denaturation	1:1000	72
β−actin	A5541 (Sigma Aldrich)	RIPA	−	1:20,000	42
Claudin-3	Ab214487 (Abcam)	RIPA	−	1:1000	20
Claudin-5	Ab15106 (Abcam)	RIPA	−	1:1000	20
ICAM-1	Ab53013 (Abcam)	RIPA	−	1:2000	80
LRP1	5A6 (Santa Cruz)	UT4	Without reduction/β−mercaptoethanol	1:200	85
NLRP3	Ab263899 (Abcam)	RIPA	−	1:1000	90
P-gp	C219 (Gene Tex)	RIPA	Without heat denaturation	1:500	180 (MDR1) 140 (MDR3)
PICALM	HPA019053 (Sigma Aldrich)	RIPA	−	1:1250	70
RAGE	Ab37647 (Abcam)	RIPA	−	1:1000	40
Occludin	Ab31721 (Abcam)	RIPA	−	1: 1000	56
Tricellulin	Ab203567 (Abcam)	RIPA	−	1: 1000	64
VCAM-1	Ab98954 (Abcam)	RIPA	−	1: 1000	100
VE-cadherin	Ab33168 (Abcam)	RIPA	−	1: 1000	115
ZO-1	Ab216880 (Abcam)	RIPA	−	1: 1000	195

## Data Availability

All relevant data are within the paper and its additional files.

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
