# Peer review of "TNF-α and IL-1β Modulate Blood-Brain Barrier Permeability and Decrease Amyloid-β Peptide Efflux in a Human Blood-Brain Barrier Model"

_ijms, 2022, doi:10.3390/ijms231810235_

Round 1

Reviewer 1 Report

The article titled, " TNF-alpha and IL-1beta modulate blood-brain barrier permeability and decrease amyloid beta peptide efflux in a human blood brain barrier model" investigates the effects of cytokine-induced neuroinflammation on several barrier properties including: barrier permeability, tight junction expression, transporter expression, and amyloid-beta clearance in an in vitro BBB model.  The manuscript was well written and the conclusions relatively match the results.  There are several improvements/suggestions/comments that could significantly improve the manuscript.

1) Please include the raw/baseline values as supplementary or in the text for control conditions in the transcellular and paracellular Pe data.  Currently they are normalized to control for each respective condition.  It would be of  interest to readers to see how PeLy changes at 37 and 4C.

2) The fact that Claudin-5 is significantly elevated at both the mRNA and protein level following TNF-alpha and the mRNA level following IL-1beta is surprising as C5 is so closely associated with barrier integrity.  In your discussion you discuss compensatory mechanisms and also the increase in delocalization/discontinuous C5 following treatment.  The manuscript would be significantly improved if the de-localization was quantified following treatment (i.e. area fraction index/ discontinuous junctions, etc.).

3) The manuscript demonstrates that several transporters expression is affected following treatment but what about their activity level.  The authors briefly discussed the utilization of inhibitors to determine their role in amyloid-beta clearance in other work.  Baseline efflux activity in control and treatment conditions would strengthen this manuscript and/or the use of inhibitors/activators  of transporters with and without treatment and their role in amyloid-beta clearance would further improve the manuscript.

4) Statement "....probably due to decreased expression of the LRP1 and BCRP which are responsible for AB peptide clearance" (Abstract line 23) should be reworded/removed.  To my point (3) the manuscript would be significantly improved if you confirmed in this model system that LRP1 and BCRP are responsible for AB clearance.

Author Response

Point 1: Please include the raw/baseline values as supplementary or in the text for control conditions in the transcellular and paracellular Pe data. Currently they are normalized to control for each respective condition. It would be of interest to readers to see how PeLy changes at 37 and 4C.

We agree with the reviewer’s comment and have therefore modified the legend in Figure 2 accordingly with the raw values of paracellular and transcellular Pe data (lines 162 - 164).

Point 2: The fact that Claudin-5 is significantly elevated at both the mRNA and protein level following TNF-alpha and the mRNA level following IL-1beta is surprising as C5 is so closely associated with barrier integrity.  In your discussion you discuss compensatory mechanisms and also the increase in delocalization/discontinuous C5 following treatment.  The manuscript would be significantly improved if the de-localization was quantified following treatment (i.e. area fraction index/ discontinuous junctions, etc.).

We appreciate the reviewer's comments, and we performed a quantification of claudin staining. As suggested, we quantified the staining area fraction of claudin-3 and claudin-5 in BLEC monolayer and we described this method in the “Materials and Methods” section (lines 753 – 759). We quantitatively demonstrated a significant increase area fraction staining of these proteins in BLECs only under the effect of treatment with TNF-α confirming that at this timepoint (24h) claudin-5 and claudin-3 did not only localize to the junction of BLECs but displayed a more diffuse expression pattern into the BLEC cytosol. Two graphs, Figures 3H and 3I, corresponding to the quantification of claudin-3 and claudin-5, respectively, were then added to the basic Figure 3. In agreement, the results (lines 267 - 275) and legend of the figure 3 has also been modified (lines 224 - 246).

Point 3: The manuscript demonstrates that several transporters expression is affected following treatment but what about their activity level.  The authors briefly discussed the utilization of inhibitors to determine their role in amyloid-beta clearance in other work.  Baseline efflux activity in control and treatment conditions would strengthen this manuscript and/or the use of inhibitors/activators of transporters with and without treatment and their role in amyloid-beta clearance would further improve the manuscript.

We thank the reviewer, and we appreciate for this comment. As suggested, the baseline apparent permeability in efflux and influx of Aß and of inulin have been added to the legend of the figure 5 (lines 339 - 342 and 344).

Then, the protein expression of LRP1 and BCRP are decreased after TNF- α or IL-1β treatment and we suggest that it can be responsible for the decrease of amyloid efflux observed. Activity levels of P-gp and BCRP in this BBB model have been extensively characterized in our several previous works using inhibitors and substrates (Cecchelli et al., 2014, Kuntz et al., 2015, Versele et al., 2020). Beyond the compensatory mechanisms of ABC transporters when some of these proteins are downregulated, it is difficult to discriminate the specific activity of a targeted ABC transporter due to the lack of specificity of the inhibitors, which impact on several members of the ABC transporters (Weidner et al., 2015 ; Moinul et al., 2022).

Regarding LRP1, we have recently shown in this BBB model and using inhibitors, that ketones bodies promote its expression that in turn increases efflux of Aβ1-40 Cy5 across BLECs (Versele et al., 2020). In these conditions, only when LRP1 is upregulated, it is possible to inhibit this receptor in order to restore Aβ transport (Versele et al., 2020).

Thus, in view of the difficulty in proving the direct involvement of these proteins in the decrease of Aß peptide efflux when their expression is decreased (Candela et al., 2015), as this is the case in this study, we ruled out to perform inhibition experiments. However, as suggested by reviewer we were careful in the discussion and reworked it. Please see lines (558-575).

Point 4: Statement "....probably due to decreased expression of the LRP1 and BCRP which are responsible for AB peptide clearance" (Abstract line 23) should be reworded/removed.  To my point (3) the manuscript would be significantly improved if you confirmed in this model system that LRP1 and BCRP are responsible for AB clearance.

As indicated above, we already investigated and demonstrated the contribution of LRP1 and efflux pumps in Aβclearance in our previous studies using this BBB model (Kuntz et al., 2015; Versele et al., 2020). However, as suggested by the reviewer, we modified the sentence in the abstract “Further, TNF-α and IL-1β act directly on BLECs by decreasing the efflux of Aβ peptide probably due to decreased expression of the LRP1 and BCRP which are responsible for Aβpeptide clearance.” and we replaced with “Further, TNF-α and IL-1β act directly on BLECs by decreasing LRP1 and BCRP protein expression as well as the specific efflux of Aβ peptide.” (lines 24 - 26).

Reviewer 2 Report

In the present manuscript, there were some minor concerns.

1. The concentration of TNF-alpha and IL-1beta was 10 ng/mL to mimic CNS inflammation. Was it similar to the serum concentration in CNS inflammation? Was it appropriate that same concentration for both cytokines was used?

2. In Figure 1B, were the immunoblotting images representative of at least three independent experiments? The quantitative data should be added.

3. In Figure 3G, why did not the authors perform staining of tricellulin?

4. Line 193, correct "(I)" to "(G)".

5. Line 199, correct "(Figure 2E)" to "(Figure 3E)".

6. In Figure 4C, the characters in label of x-axis were funny.

7. Discussion might be too long.

8. In References, number 1 and 69 were same paper. It should be corrected and renumbered. 

Author Response

#Reviewer 2

Point 1: The concentration of TNF-alpha and IL-1beta was 10 ng/mL to mimic CNS inflammation. Was it similar to the serum concentration in CNS inflammation? Was it appropriate that same concentration for both cytokines was used?

Unfortunately, there is not much data in in vitro studies distinguishing peripheral from central inflammation. Hence, the used concentrations of 10 ng/mL for TNF- α or IL-1β to mimic inflammation were selected based on: Camire et al., 2015; Lopez-Ramirez et al., 2013; Maeda et al., 2021; Ni et al., 2017; Rochfort et al., 2014; Zhang et al., 2019. These concentrations are one of the most widely used in in vitro BBB models to mimic brain inflammation, despite serum concentration described to be in the pg/mL range (Goikolea et al., 2021). However, in vivo data about pro-inflammatory cytokine levels in serum are often discordant. Indeed, results differ depending on many parameters, including the technique used, the stage of Alzheimer's disease patients, age, gender and even the patient's inflammatory comorbidity (Brosseron et al., 2014; Lee et al., 2013; Swardfager et al., 2010).       

Then, the use of same concentration between TNF- α or IL-1β in our study is appropriate from our point of view. Indeed, Grammas and Ovase have reported that the secretion of these two pro-inflammatory cytokines are equivalent (almost 30 pg/mL) in brain microvessel of AD patients (Grammas & Ovase, 2001). This suggests that the BBB and CNS are exposed to equivalent concentrations of TNF- α or IL-1β.

For all these reasons, we have chosen these concentrations that are able to induce an inflammatory response in BLECs and validate the use of our BBB model to study the effect of cytokines on BBB functions.

Some of these s reference and these explanations have been added in the manuscript (lines 379-385)

Point 2: In Figure 1B, were the immunoblotting images representative of at least three independent experiments? The quantitative data should be added.

We thank the reviewer for this comment and we totally agree with this point raised. However, no quantitative data can be obtained for VCAM-1, NLRP3 and ICAM-1 in control condition because there is no signal (please see Figure 1B). Therefore, to reply to the reviewer’s comment, we added the figure 1C corresponding to the quantification of the protein levels of VCAM-1, NLRP3 and ICAM-1 following IL-1β treatment compared to TNF-α treatment. The legend of Figure 1 has been modified accordingly (lines 110 – 117) and the p-value has been added in the text (lines 131 - 132). Hope that this modification will be appreciated by the reviewer.

Point 3: In Figure 3G, why did not the authors perform staining of tricellulin?

Thank you for pointing out this shortcoming but we tried to perform tricellulin staining in BLECs but it did not work despite the use of different immunostaining protocols with the Abcam reference (Ab203567). As no alternative anti-tricellulin antibodies were available in the commerce for IF of human tissues/cells, we were not able to show a pattern of tricellulin staining in BLECs.

Point 4: Line 193, correct "(I)" to "(G)".

We thank the reviewer for pointing out this typing error which has been corrected (line 222).

Point 5: Line 199, correct "(Figure 2E)" to "(Figure 3E)".

We thank again the reviewer for pointing out this typing error which has been changed (line 250).

Point 6: In Figure 4C, the characters in label of x-axis were funny.

Unfortunately, we have not identified the problem in the figure. It could be that the problem is due to a compatibility problem with the image format. We have therefore uploaded the image again, hoping to have solved the problem that only this reviewer pointed out.

Point 7: Discussion might be too long.

We thank again the reviewer and we have therefore reworked and cut some parts of the discussion. Please see the revised discussion.

In addition, we moved part discussion to introduction (lines 85-88). 

i

Point 8: In References, number 1 and 69 were same paper. It should be corrected and renumbered.

We again appreciate the highlighting of this error which has been corrected.

Reviewer 3 Report

This is a very well described study and clearly presented. The details in the figures are very well delivered. The statistical details are well described, and interpretation of the data is described in an unbiased manner. It appears the literature is generally well covered in relation to the details of this study. This manuscript is informative and will help advance the field in addressing the factors like the inflammatory cytokines studied.

This will likely lead others to investigate similar types of studies in addressing the role of compounds affecting the BBB. In is interesting that the authors mentioned the CNS side and the other side of the BBB might respond differently to cytokines or even other factors. This is an interesting concept to ponder.

The in vitro experimental model being used in this study has so much potential it is model readers not familiar with would enjoy learning about and I am sure there are more studies in the works with the use of this model of the human BBB. I thoroughly enjoyed reading this manuscript.  I do not have an edits or minor corrections for the authors. I am not sure of the format for this journal, but the Conclusions after the Methods sectioned seemed odd. Would it not be better right after the Discussion section?

Author Response

#Reviewer 3

Point 1: I am not sure of the format for this journal, but the Conclusions after the Methods sectioned seemed odd. Would it not be better right after the Discussion section?

We thank the reviewer for the comment. This is a recommended formatting and not an imposed one. Thus, in line with your comment and in order to facilitate the reading of the article, we have moved the conclusion paragraph to the end of the discussion as you suggested (lines 576 - 634). 

Round 2

Reviewer 1 Report

The authors addressed my comments appropriately.